# Structural Evolution and Transitions of Mechanisms in Creep Deformation of Nanocrystalline FeCrAl Alloys

**DOI:** 10.3390/nano13040631

**Published:** 2023-02-05

**Authors:** Huan Yao, Tianzhou Ye, Pengfei Wang, Junmei Wu, Jing Zhang, Ping Chen

**Affiliations:** 1School of Aerospace, Xi’an Jiaotong University, No.28, Xianning West Road, Xi’an 710049, China; 2School of Nuclear Science and Technology, Xi’an Jiaotong University, No.28, Xianning West Road, Xi’an 710049, China; 3Nuclear Power Institute of China, No.328 Huayang Changshun Avenue Section 1, Chengdu 610213, China

**Keywords:** creep deformation, mechanism, nanocrystalline, molecular dynamics

## Abstract

FeCrAl alloys have been suggested as one of the most promising fuel cladding materials for the development of accident tolerance fuel. Creep is one of the important mechanical properties of the FeCrAl alloys used as fuel claddings under high temperature conditions. This work aims to elucidate the deformation feature and underlying mechanism during the creep process of nanocrystalline FeCrAl alloys using atomistic simulations. The creep curves at different conditions are simulated for FeCrAl alloys with grain sizes (GS) of 5.6–40 nm, and the dependence of creep on temperature, stress and GS are analyzed. The transitions of the mechanisms are analyzed by stress and GS exponents firstly, and further checked not only from microstructural evidence, but also from a vital comparison of activation energies for creep and diffusion. Under low stress conditions, grain boundary (GB) diffusion contributes more to the overall creep deformation than lattice diffusion does for the alloy with small GSs. However, for the alloy with larger GSs, lattice diffusion controls creep. Additionally, a high temperature helps the transition of diffusional creep from the GB to the dominant lattice. Under medium- and high-stress conditions, GB slip and dislocation motion begin to control the creep mechanism. The amount of GB slip increases with the temperature, or decreases with GS. GS and temperature also have an impact on the dislocation behavior. The higher the temperature or the smaller the GS is, the smaller the stress at which the dislocation motion begins to affect creep.

## 1. Introduction

Nuclear power has been recommended as a sustainable and cleaner way to produce electricity [1]. The high-power density of the core makes nuclear power cost-effective, but it is also susceptible to severe accidents. How to intensify the safety of nuclear systems has been more of a concern for the international nuclear community since the 2011 Fukushima nuclear accident. The concept of accident-tolerant fuel (ATF) has been proposed and is expected to slow the core degradation process during severe accidents and to maintain or upgrade the fuel performance under normal operations [2]. In the development of ATF, the fuel cladding material is key to resisting high temperature, intense radiation and water corrosion/steam oxidation and to ensuring the structural integrity of fuel elements under severe accident conditions. It is widely acknowledged that the Zirconium-water reaction plays a vital role in fuel cladding failure under a loss of coolant accident (LOCA). As one of the alternatives to existing Zr-based cladding, FeCrAl alloys have a superior resistance to hot steam under a LOCA, generating less combustible and explosive hydrogen gas [3]. Several researchers have also noted that FeCrAl cladding can improve the margin of safety and provide more time to cope with further essential risks after the LOCA [4,5]. In this regard, FeCrAl alloys have attracted great attention as promising candidates as an ATF cladding material for light water reactors. In view of their superior resistance to corrosion and oxidation in high-temperature water/steam environments [6,7,8,9], these alloys are also supposed to be suitable for the coating material on traditional zircaloy cladding surfaces [10].

Creep refers to a solid material that experiences a slow increase in strain over time at high temperatures under long-term stress circumstances. It is one of the influential mechanical properties to ensure the safety and economy of solid materials in high-temperature applications. The fuel cladding serves as the first physical barrier in nuclear reactors, so maintaining the integrity of the cladding is necessary and very important. As the candidate for ATF cladding materials, a nuclear-grade FeCrAl alloy is subjected to a high-pressure and high-temperature water/steam atmosphere. Therefore, in-depth investigations of the characteristics and mechanisms of the FeCrAl alloy’s creep deformation are very significant for its design and application to ATF cladding. Some studies on the FeCrAl alloy’s creep which were carried out recently are reviewed as follows.

In recent years, experimental research has been conducted on the creep behavior of oxide-dispersion-strengthened (ODS) FeCrAl alloys for their excellent creep strength. Several studies have shown that the creep deformation of ODS FeCrAl alloys is mainly governed by a dislocation motion at high stress, and then the grain boundary (GB)-assisted deformation becomes prominent with the decreasing stress [11,12,13]. Ukai et al. [14] carried out high-temperature ring-creep tests for ODS FeCrAl cladding to evaluate the creep mechanism under severe accident conditions. Creep data suggested that the GB slip controls the creep at low stress, and then the dislocation climb becomes the dominant mechanism when the stress is above the stress at which the dislocations detach. Kamikawa et al. [15] found that the creep mechanism is dominated by the GB slip and diffusion creep under low-stress conditions, followed by the dislocation climb under higher stresses. Furthermore, the creep behavior and underlying mechanisms of the ODS FeCrAl alloy under different temperatures are reported in the literature. Sun et al. [16] performed a work recently on the strain rate jump of nanolaminate FeCrAl alloys from room temperature to 350 °C. It was pointed out that GB diffusion is the key factor to controlling the creep rate. Dryepondt et al. [17] pointed out that the creep deformation is most likely due to a GB slip for the ODS FeCrAl alloys at 800 °C. In contrast, the dislocation climb was identified as the primary creep mechanism at lower temperatures from 400 to 450 °C [18]. Additionally, the studies on the behavior of a burst of FeCrAl cladding have also shown that the dislocation climb is the major rate-limiting factor at temperatures from 480 to 650 °C [19,20].

The subject related to the creep properties of other engineering materials has also been studied over recent decades. Katouzian et al. [21] proposed a micromechanical model to predict the creep performance of viscoelastic materials, and an excellent agreement has been shown between their experimental and computed results. Ma et al. [22] performed the nanoindentation creep tests on metallic glassy film and found that the indenter size has a strong effect on creep resistance. The creep deformation becomes more apparent with smaller indentation sizes. Karbasi et al. [23] reported that a nano-structured Co/Pb composite exhibits an enhanced tensile and creep strength compared to the pure Pb samples. Mokhtari et al. [24] investigated the creep behaviors of nanocomposite fibers and found that the creep resistance can be improved with the addition of nanoparticles. Nanocrystalline (NC) materials appear to have a new and generally superior physical-chemical property over traditional coarse-grained materials [25,26]. Within the grain size (GS) of 1–100 nm, NC materials have a potential application in high-temperature environments for enhanced mechanical strength due to their grain refinement. In this case, it is worth taking some effort to evaluate the high-temperature creep performance and deformation features of NC FeCrAl alloys for their potential engineering applications. However, the present experimental research on NC materials is insufficient because of the high costs of preparation and experimentation. The available experimental literatures regarding creep performance mainly focus on micro-scale materials. Fortunately, theoretical analysis and computer simulations have proven to be very economical and powerful tools in the development of new materials, particularly at a nanoscale. Molecular dynamic (MD) simulation is a computational simulation method for solving multi-body motion at the atomic and molecular levels, and plays an increasingly important role to understand the feature of structural evolution and the corresponding mechanism during creep deformation.

The MD method has so far been utilized to reveal and explain the deformation behavior and related mechanism for the creep process of NC metals such as Cu, Mg and Ni, as well as NC alloys such as Mg-Y, Ni-Zr, Ti-Al and Fe-Ni-Cr alloys. Due to the limitations of computational time and spatial scales, the strain rate calculated by the MD simulation is often on the order of 10^5^ s^−1^ or more, which is higher than the creep experiments. Hence, the researchers pay more attention to the variation of the related mechanisms with temperature, stress and GS rather than the strain rate itself. Earlier, a general model for the stress-driven transition of the creep mechanism in NC metals with stress was built in the work of Wang et al. [27]. It was pointed out that the dominant mechanism shifts from GB diffusion to GB slip, and then dislocation creep with the increase of stress. In another study, they further analyzed the grain size effect of creep in NC metals [28]. Similarly, the creep mechanism of NC Ni is discovered to change from the synergy of diffusion creep and GB slip to a dislocation-motion-governed creep with increasing stress and temperature and decreasing grain size [29]. Meraj et al. [30] pointed out that at the beginning of the steady-state creep of NC Ni, the creep process is dominated first by GB diffusion and then shear diffusion. They also found that the pre-existing nano-crack is helpful to improve the creep strength of NC metals [31]. More recently, a few studies have focused on NC alloys’ deformation features and their related mechanisms during the creep process. Bhatia et al. [32] simulated the creep process in NC Mg and Mg-Y alloys with the GSs of 5 nm and 10 nm. In the case of NC Mg, GB diffusion plays a vital role in the overall creep behavior at low temperatures, while lattice diffusion becomes predominant when the temperature exceeds 573 K. It has also shown that GB slip/rotation dominates the process of creep deformation in NC Mg-Y alloys. Zhao et al. [33] found that the deformation behavior in the steady creep stage of NC Ti-Al alloys is governed by GB diffusion at low stress, and then dislocation motion at higher stress conditions. In contrast, lattice diffusion and GB diffusion are considered to be predominant in the third creep stage. It is reported by Pal et al. [34] that the primary and secondary creep stages of the NC Ni and Fe-Ni-Cr alloy nano-joint are dominated by GB diffusion and dislocation creep. Furthermore, Pal et al. [35] studied the impact of Zr addition on the creep behavior of NC Ni and found that the creep properties can be improved with distributed Zr atoms at GBs. The implication of segregating Zr to Ni GBs on the deformation mechanism for the process of bending creep is assayed in their following work [36].

The above-mentioned work reveals that the MD method can be successfully used to predict the creep performance and deformation features for NC materials. The researchers can better comprehend the relationship between the creep behavior and related mechanism with the use of MD. It is noted that an atomistic simulation for the creep deformation, structural evolution and the underlying mechanisms of an FeCrAl alloy is limited, although experimental research on an FeCrAl alloy at a micron scale has been widely released in recent years. In our previous work, the structural changes and deformation features of NC FeCrAl alloys during creep have been preliminarily evaluated with the help of the MD code LAMMPS [37].

The overall objectives of the current study are to provide an atomistic understanding of the structural evolution and underlying mechanisms for the creep process of an NC FeCrAl alloy by using the MD method. The creep curves and corresponding creep rate at different temperatures and stress conditions were simulated for FeCrAl samples with GSs from 5.6 to 40 nm, to evaluate the dependence of the creep on temperature, stress and grain size. Furthermore, the transitions of major mechanisms affected by these factors are explored.

## 2. Simulation Model and Methodology

### 2.1. NC FeCrAl Models

The available research found that FeCrAl alloys exist as an Fe-based solid solution combined with the substitutional Cr and Al solutes due to the lower formation energies of vacancy [38,39]. It is proposed that 13Cr and 5Al could offer better processing, thermophysical and mechanical properties for FeCrAl fuel cladding applications [9]. Thus, by randomly doping with Cr and Al substitutional atoms in the bcc α-Fe matrix, the three-dimensional single-crystal Fe-13Cr-5Al models are constructed firstly. Then, the Voronoi tessellation method [40] is applied to establish NC Fe-13Cr-5Al simulation models in the present work. Figure 1 shows the established three-dimensional NC FeCrAl models with GSs of 5.6, 10, 16, 20, 30 and 40 nm, respectively. To identify GBs and intragranular regions, the atoms are classified by crystal structure in crystalline systems using the common neighbor analysis (CNA) method [41]. Table 1 shows the parameters of the models with different GSs.

### 2.2. Simulation Method and Conditions

In this study, all the calculations are carried out with LAMMPS Molecular Dynamics Simulator referring to the literature [42]. To obtain a good initial structure, energy minimization was performed on all models by the conjugate gradient method. Then, all samples were equilibrated with NVT (constant the total number of atoms, volume and temperature) ensemble at target operating temperatures from 600 to 1500 K for 300–15,000 ps and then relaxed with NPT (constant of the total number of atoms, pressure and temperature) ensemble for another 300–15,000 ps. These two processes are performed to eliminate the residual and thermal stresses in a crystalline system for model initialization. Then the creep simulations were conducted for 300–20,000 ps in the NPT ensemble, under different applied stresses in the range of 0.3–3.0 GPa along the *Y*-axis, while ensuring that there was zero applied stress in the other two directions. The time step was taken as 1 fs and periodic boundary conditions were applied to the opposite surfaces in three axis directions. Atomic arrangement snapshots capturing, Wigner–Seitz defect analysis, Common neighbor analysis (CNA) and dislocation analysis were carried out and visualized using OVITO [43]. CNA is a helpful tool to identify the local crystal structure around an atom [41] in MD simulation. The evolution of the crystal structure in NC FeCrAl samples was analyzed based on CNA. Wigner–Seitz defect analysis was executed to view the spatial distribution of point defects and count their number under different stages of creep deformation. By comparing with the perfect lattice, where every site is occupied by exactly one atom, this analysis effectively determines the location of vacancy or interstitial defects. In this method, the lattice position is treated as a vacancy if no atoms occupy it. In addition, if one site is occupied by more than one atom, the identified defect is interstitial. In order to check the variation of dislocation density and distribution with creep time, dislocation analysis is also carried out at different loading conditions.

### 2.3. Methods for Analysis and Verification of Creep Mechanism

The strain rate data in the steady-state creep stage are analyzed to evaluate the stress exponent n, GS exponent p and the activation energy of creep Q_c_, which are parameters suggestive of the underlying mechanism. Typically, a general constitutive equation of Equation (1) relating the creep rate $\dot{\varepsilon }$ with stress σ, grain size d and temperature T through n, p and Q_c_, respectively, has been proposed by Mukherjee and Bird et al. [44]
(1)$$\dot{\varepsilon }=\frac{{\mathrm{AD}}_{0}\mathrm{Gb}}{k_{B}T}{\left(\frac{b}{d}\right)}^{p}{\left(\frac{\sigma }{G}\right)}^{n}\exp\left(-\frac{Q_{c}}{k_{B}T}\right)$$
where A is a constant, b, G and D_0_, namely, are the Burgers vector, shear modulus (Pa) and atomic diffusion coefficient (m^2^/s), and k_B_ is Boltzmann’s constant. Referring to Equation (1), n, p and Q_c_ can be defined as follows:(2)$$n=\frac{\partial \log\dot{\varepsilon }}{\partial \log\sigma }$$
(3)$$p=\frac{\partial \log\dot{\varepsilon }}{\partial \mathrm{logd}}$$
(4)$$Q_{c}=\frac{\partial \ln\dot{\varepsilon }}{\partial \left(\frac{1}{k_{B}T}\right)}$$

The creep deformation mechanisms transform mainly with the variations of n and p. In relation to n = 1, the deformation mechanism is known as diffusion creep within lattice when p = 2 [45] or in the GB region when p = 3 [46]. The creep process is controlled by GB slip with respect to n = 2 [47,48]. For n ≥ 3, the creep mechanism is considered to be dislocation motion, which occurs via dislocation slip and dislocation climb [49,50,51,52].

In this study, the creep process of FeCrAl samples with GSs of 5.6–40 nm are simulated for a range of temperatures and stresses, to obtain their creep strain curves. The creep curve in the steady creep stage is linearly fitted to obtain the creep rate under different conditions. Referring to Equations (2)–(4), the stress, GS and temperature dependence of creep rate can be quantitatively evaluated by n, p and Q_c_, respectively, which are key parameters indicative of the underlying mechanism. Based on the variations of n and p, the transitions of creep mechanisms in NC FeCrAl are preliminarily analyzed and checked from microstructural evidence. Furthermore, a detailed comparison of the activation energies for creep Q_c_ and diffusion Q_d_ is performed to correlate the creep process with atomic diffusion over a range of stress and GS.

## 3. Results and Discussion

### 3.1. Analysis of Simulated Creep Curves and Related Creep Rate

In this section, the creep behavior at different conditions is simulated for Fe-13Cr-5Al alloys with GSs of 5.6–40 nm. In general, the creep strain curves are composed of the three typical stages, primary creep, steady-state creep, and tertiary creep. At the primary creep stage, the strain rate gradually decreases and reaches a constant value as the deformation process comes into the steady-state creep. In the end, the tertiary creep stage’s strain rate quickly rises.

#### 3.1.1. Interatomic Potential Effect

As we know, interatomic potential provides information pertinent to the fundamental aspects of the interaction between atoms. The accuracy of the interatomic potential will directly affect the accuracy of the MD simulation results. Concerning the FeCrAl ternary system, interatomic potentials have been developed with different approaches in the literature over the past few years. Liu et al. [53] developed a potential (Liu-potential) by using the particle swarm optimization (PSO) method. It has great potential for predicting the phase stability region and mechanical properties. The predictions from the Liu-potential are coincident with the experimental and first principle calculated results. However, a few disadvantages exist which are related to the formation and migration energies of point defects. Liao et al. [54] recently developed a potential (Liao-potential) through the Finnis–Sinclair formulism, with the emphasis on the defect and thermodynamic properties. The potential parameters were checked by comparing them with a set of experimental data. Figure 2 shows the dependence of the creep strain on the temperature and stress calculated by the two different potentials. All creep curves exhibit similar characteristics, which are a short primary creep and a relatively longer steady-state creep. It can also be seen that the creep strains calculated by the Liao-potential are higher than those calculated by the Liu-potential. The strain rate (the slope of the creep curve), calculated by the Liao-potential, greatly increases with temperature, while the Liu-potential underestimates this effect. Meanwhile, the sensitivity of the creep rates to stress obtained from the two potentials are in agreement generally. It is known that creep deformation is thermally activated and is strongly influenced by temperature. Thus, it is suggested that the Liao-potential is more appropriate for simulations of structural evolution and deformation features during thermal creep in FeCrAl alloys.

#### 3.1.2. Tensile Direction Effect

Figure 3 provides the simulated creep curves for a 5.6 nm grain size Fe-13Cr-5Al sample at 1200 K and 0.5 GPa in different tensile directions. One can find that the creep strain is obviously different in different stress loading directions. However, the steady creep rate does not change a lot with the stress loading directions. It suggests that the steady-state deformation features may be slightly correlated with the tensile direction. This can be elucidated as follows. In this work, the models of NC FeCrAl are comprised of grains having random positions and crystal orientations. During the creep process, the deformation behavior in three directions is bound to be interconnected and coordinated with each other. The creep rate in each direction is generated by the superposition of the creep deformation in all directions. So, the tensile direction has only a little impact on the creep rate and the following results are based on the simulation work with the tensile direction in the *Y*-axis direction.

#### 3.1.3. Stress and Temperature Effects

Figure 4a provides the simulated creep curves for a 5.6-nm grain size Fe-13Cr-5Al sample under different temperatures at 0.5 GPa. Under this low-stress condition, all the creep curves under different temperatures show a brief primary creep followed by an extensive steady-state creep. It is also shown that the creep rate increases greatly as the temperature rises and the higher the temperature, the faster the increase rate becomes. This is due to the fact that a higher temperature accelerates the point-defect diffusion as well as the thermally activated processes such as the dislocation climb and glide. Hence, the creep deformation induced by the diffusion creep and dislocation motion is increased with the rising temperature.

Figure 4b provides the simulated creep curves for a 5.6-nm grain size Fe-13Cr-5Al sample at different stress levels for a fixed temperature of 1200 K. As the stress increases, the creep rate grows rapidly, especially at stresses above 1.5 GPa. It can also be seen that a tertiary creep appears at stresses over 1.8 GPa in our simulation time period. This is because high stress not only helps to overcome the energy barrier for dislocation motions and the formation and diffusion of point defects, but also to promote GB motions including GB slip and migration.

#### 3.1.4. GS Effect

Figure 5 provides the simulated creep curves for Fe-13Cr-5Al alloys with grain sizes from 5.6 to 40 nm at 0.5 GPa for 1200 K. In the figure, the strain rate decreases quickly with GS, varying from 5.6 nm to 12 nm, after that, it changes slowly. This can be interpreted as follows. As we know, GBs are exactly the interfaces that separate crystal grains with different orientations, which are highly responsible for the relative motions between grains, such as GB slip and grain rotation. With respect to NC materials, GBs serve as important sources and sinks of dislocations and are the favorable channel for the diffusion of point defects at sufficiently small GSs [55]. It can be seen from Table 1 that the fraction of GB atoms significantly decreases with GS increasing, thus leading to a continuous decrease in the creep rate. This accords with the relationship between the creep rate and the GS proposed in Equation (1).

### 3.2. Analysis of Steady-State Creep Stress and Grain Size Exponents

As described in Section 2.3, the dominating mechanism can be preliminarily determined by steady-state creep stress exponent n and grain size exponent p. This section attempts to assay the transitions of major mechanisms in FeCrAl alloys in terms of the variations of n and p. Based on Equations (2) and (3), n and p can be determined by drawing double-logarithmic plots for the steady-state creep rate with respect to stress and grain size, respectively, plotted in Figure 6, Figure 7, Figure 8 and Figure 9. From Figure 4b, the variation of the creep rate is greatly related to the stress, so the double-logarithmic curves for the steady-state creep rate versus stress are fitted piecewise and linearly in two or three segments with the different stress exponents of n. Additionally, the double-logarithmic curves for the steady-state creep rate versus grain size are fitted piecewise and linearly in two segments with different GS exponents of p, because the impact of GS on the creep rate is getting smaller and smaller after the GS is larger than 12 nm, as shown in Figure 5.

#### 3.2.1. Variation of Stress Exponent n

Figure 6 provides the double-logarithmic curves for the steady-state creep rate against stress at different temperatures from 600 to 1400 K. It is found that n is gradually increased with increasing stress and temperature. At the lower temperatures (less than 1000 K), n is about 1.3 at low stress and is nearly 2.0 at medium stress. Then, n becomes highly temperature-dependent and increases greatly to more than 3.0 under higher stress. As the temperature rises up to 1200 K and above, n is close to 1.4 under low stress and varies between 2.4 and 3.3 when the stress ranges from 0.8–1.5 GPa, and then it increases to be larger than 5.0 when stress exceeds 1.5 GPa. This indicates that the temperature has an impact on the variation of n, as well as the turning point of the stress of the three linear segments. The turning point of stress between the medium-stress exponent and the high-stress exponent drops from 2.2 to 1.5 GPa with the rising temperature. These results suggest that both temperature and stress have an obvious effect on the stress exponent and, potentially, the variations of temperature and stress will lead to the transition of the deformation mechanism in an NC FeCrAl alloy.

The impact of the GS on n at different temperatures is also analyzed and presented in Figure 7 and Figure 8. One can find that at the low temperature of 800 K, the GS has little influence on the variation of the stress exponent and the turning point of stress. Meanwhile, the GS has an impact on the turning point of stress between the medium-stress exponent and the high-stress exponent at high temperatures. Under the smaller GSs, the steady-state creep rate has a linear fitting to stress in three stages, with turning points of 0.8 GPa and 1.5 GPa. However, when the GS increases to above 12 nm, the double-logarithmic curves for the steady-state creep rate against stress are more properly fitted in two linear segments, taking 1.8 GPa as the turning point. It is well recognized that the variation of n corresponds to the transition of the creep mechanism. Therefore, we can speculate that the variation of the GS may also affect the transitions of major mechanisms in FeCrAl alloys within the present GS range. In addition, the values of n decrease slightly with the increasing GS. This indicates that the dependence of the creep behavior on the applied stress is higher under small GS conditions.

#### 3.2.2. Variation of GS Exponent p

Figure 9 shows the double-logarithmic curves for the steady-state creep rate against the grain size under different temperatures and three stress levels. Two segments regarding different fitted p are divided with the turning point of 12 nm. It is evident that the GS exponent p decreases greatly as GS rises at different stresses and temperatures. This suggests that the change in GS will lead to the transition of the underlying mechanism. It is also shown that the GS exponent p for NC FeCrAl models has little difference and is less than 1 when GS exceeds 12 nm. However, for the NC FeCrAl models with smaller GSs, the GS exponents of p may increase from about 2 to 3 with the increasing stress and temperature. According to Section 2.3, this change in p from about 2 to 3 corresponds to the transition of the creep mechanism from lattice diffusion to GB diffusion, in consideration of GS’s effect.

### 3.3. Microstructural Evidence for the Transitions of Creep Mechanisms

The creep mechanism’s transition is originated from the competition among different thermally activated and stress-driven processes, which include GB and lattice diffusion, GB slip and dislocation motion. As described in Section 2.3, generally, the dominant creep mechanism varies with varying values of n and p. So, based on the results of Figure 6, Figure 7, Figure 8 and Figure 9, the transitions of the creep mechanism for NC Fe-13Cr-5Al with stress, temperature and GS are speculated and summarized in Table 2. To better understand the results of Table 2, we will elucidate the transitions of the creep mechanism with providing the microstructure evolution of NC Fe-13Cr-5Al at different conditions during the creep process.

As we know, lattice diffusion and GB diffusion are due to the diffusion of point defects within the lattice or along the GBs. GB slip creep refers to the relative movement of neighboring grains along GBs under the action of shear stress. Dislocation creep arises from the two competing deformation processes of dislocation glide and climb, controlled by diffusion. Based on this, Wigner–Seitz defect analysis, coordination analysis and an atomic strain calculation have been performed to capture the diffusion creep in NC FeCrAl, shown in Figure 10, Figure 11 and Figure 12. Representative CNA atomic snapshots and displacement vector distribution are given to observe the GB slip behavior, shown in Figure 13, Figure 14, Figure 15 and Figure 16. Dislocation analysis is conducted to provide the microscopic evidence for the dislocation motions, shown in Figure 17, Figure 18 and Figure 19.

#### 3.3.1. Diffusional Creep

Under low stress conditions, n is found to be 1.0–1.5 and diffusion-driven mechanisms are likely to be dominant. To capture the diffusional creep behavior in NC FeCrAl under this condition, Figure 10 shows the variation of the vacancies’ concentration with time under different temperatures and GS conditions with a stress of 0.5 GPa. The vacancy concentration increases with temperature and decreases with GS. It is similar to the related creep curves of Figure 4a and Figure 5. It indicates that there is a significant correlation between creep deformation and the formed vacancies due to the diffusion effect as the creep process progresses. Samples snapshots can also visualize the density and distribution of vacancies and interstitials of the studied samples at different times. So, Figure 10 gives the samples snapshots of NC Fe-13Cr-5Al at 300 ps under three GS and three temperature conditions as well. It is observed that the density of the point defects (vacancies and interstitials) is much higher in GBs than that inside the grains. This is because the atomic coordination number of the samples at the beginning of the creep simulation is lower in GBs compared to the bulk of the atoms within the grains, as shown in Figure 11. Thus, GB atoms are more likely to diffuse and migrate under the same loading condition. We also observe that the number of coordination atoms within the grains decreases with the increasing temperature and GS, which further leads to the increase of vacancies and interstitials. In this case, there may be a shift of the dominant mechanism from GB diffusion to lattice diffusion with the increase in temperature and GS.

It is known that the diffusion process is accompanied by the generation of atomic shear strain. To discern whether lattice or GB diffusion dominates the deformation process of NC FeCrAl samples at low stress, the fraction of atomic shear strain in GBs and grain interiors to the overall shear strain of the samples was calculated and compared. Figure 12 provides the plots of the fraction of atomic shear strain in the grain interiors to the overall shear strain vs. GS under different temperatures at 0.5 GPa. It is shown that the fraction of the atomic shear strain in grain interiors increases continuously with the increase in temperature and GS. When the fraction of atomic shear strain in the grain interiors exceeds 50%, the dominant creep mechanism could be recognized as lattice diffusion. From Figure 12, one can find the shift from GB diffusion to lattice diffusion is related to GS and temperature. In summary, GB diffusion (namely Coble creep) is dominant for samples with small GSs, while lattice diffusion (namely Nabarro-Herring creep) is dominant for samples with large GSs. For example, for the FeCrAl sample with a 5.6-nm grain size, the fraction of atomic shear strain is under 50% within the whole studied temperature range (600–1400 K), meaning that GB diffusion is dominant. For NC FeCrAl with a GS above 18 nm, the fraction of atomic shear strain exceeds 50% within the whole studied temperature range (600–1400 K) and so the dominant creep mechanism should be lattice diffusion. For NC FeCrAl with a GS of 10 nm, the fraction of atomic shear strain in the grain interiors is about 37.1% at 600 K and increases to above 50% when the temperature exceeds 1200 K. This implies that a high temperature helps the shift of the dominant mechanism from GB diffusion to lattice diffusion.

#### 3.3.2. GB Slip Creep

Under medium stress, the dominating mechanism is considered as GB slip regarding n ≈ 2. To elucidate through the visible evidence of the major role of GB slip in creep deformation, five groups of atoms across GBs in different orientations were selected and marked as A–E. The selected atoms in one group were on a straight line at τ = 0 ps, then their locations were recorded to capture the relative slip motion between grains on GBs under different creep conditions. Figure 13a–d show the atomic arrangement snapshots colored by the crystal structure through CNA analysis for the 10-nm grain size FeCrAl sample at different creep times at 1.5 GPa for 1200 K. It is observed that a GB slip movement does not exist between different grains, and it becomes more evident with the progress of the creep process. In the initial period of 0–5000 ps, GB slip occurs only in the atoms of group A, shown in Figure 13b, and four groups of atoms have experienced a GB sliding motion at τ = 15,000 ps, from Figure 13d. The vector displacements of atoms of the samples can visualize the GB slip plane between the grains. So, Figure 13e gives the displacement vector fields of NC Fe-13Cr-5Al as well. It is observed that the slip system alongside the tensile direction of the *Y*-axis is remarkably activated in the vicinity of the GB regions. This implies that the relative sliding of the grain boundaries occurs predominantly in the direction of the maximum resolved shear stress.

The impact of applied stress on the GB slip deformation is also investigated. Figure 14a–d present atomic snapshots colored by a crystal structure through CNA analysis at 10,000–15,000 ps under 1200 K and four different stress conditions. Figure 14a shows that at the stress of 0.5 GPa, the atoms in the A–E groups keep on a straight line even as the creep process proceeds at τ = 15,000 ps and no obvious atomic displacement is observed along the GBs. This result indicates that no obvious GB slip occurs during the creep deformation process under this low-stress condition. However, when stress is above 0.5 GPa, obvious GB slip deformation is observed and the GB slip becomes more significant with the increase of stress, as shown in Figure 14b–d. At the stresses of 1.5 and 2.0 GPa, four groups of atoms have undergone GB slip motion. These results show that the level of stress applied plays a key role in the process of GB slip, which occurs mainly at medium and high stress levels. Moreover, the lattice distortion in the grains and GBs becomes larger at higher stress levels.

The variations of the GB slip rate with GS and temperature are also checked at a medium stress. Figure 15 presents atomic snapshots colored by a crystal structure through CNA analysis at different temperatures for 1.5 GPa. It is found that the GB slip behavior becomes more apparent and the GB slip rate increases with temperature. This is because, at high temperatures, polycrystal’s binding strength at the grain boundaries is significantly weakened. GB slip only occurs in the atoms of group C at 800 K, while four groups of atoms are engaged in the GB sliding motion at 1200 K. This suggests that at medium stress, GB slip deformation does occur for different temperatures, and their amount increases greatly as the temperature rises.

Figure 16 shows atomic arrangement snapshots colored by a crystal structure through CNA analysis for an Fe-13Cr-5Al sample with different GSs from 5.6 to 20 nm. To capture the relative slip motion of the adjacent grains, 4–8 groups of atoms across GBs in different orientations were selected and marked as A–H at the beginning of the creep simulation, shown in Figure 16a–c. It can be observed that the number of active slip planes and the GB slip rate decrease drastically with the increasing GS. For the 5.6-nm grain size sample, four groups of atoms experienced obvious GB slip motion at τ = 1500 ps. Meanwhile, GB slip occurs only in group E atoms at the larger GS of 20 nm even at 20,000 ps. This can be explained by the fact that the GB atoms are greatly decreased with the increasing GS. From Figure 13, Figure 14, Figure 15 and Figure 16, we can believe with great certainty that GB slip plays an important role in the overall creep deformation for medium stress. The GB slip rate increases with temperature or decreases with the increase in grain size.

#### 3.3.3. Dislocation Creep

Under high-stress conditions, n ≥ 3 suggests that the dominating mechanism for the creep process of NC FeCrAl should be dislocation creep combined with the GB slip. To verify it, the variation of the total dislocation density with the time for NC FeCrAl with a GS of 10 nm at different stresses and temperatures is counted, as shown in Figure 17. It can be observed that the dislocation density of samples does not change significantly with time at moderate and low stresses, indicating that the dislocation motion is not the major mechanism. In contrast, at a higher stress level, the total dislocation density decreases gradually with the progress of the creep process and the decrease rate of the total dislocation density increases with the rising stress. This suggests that dislocation plays a role in the creep process under high stress conditions. By considering this together with the result shown in Figure 14d, it can be seen that the GB slip also occurs at higher stress levels, thus the dislocation creep combined with the GB slip are recognized as the dominant creep mechanism under high-stress conditions. It is also observed that the stress at which the dislocation density begins to decline varies with the temperature, and the turning point of stress decreases from about 2.2 to 1.5 GPa as the temperature rises from 600 to 1400 K. This follows the turning point of stress from the medium-stress exponent to the high-stress exponent shown in Figure 6.

To identify the dislocation motion generated, whether in GBs or grain interiors mainly, the volume fractions of the GB dislocations vs. the time at different temperatures and stresses are computed. Figure 18 gives the volume fractions of the GB dislocations vs. the time for a 10-nm grain size Fe-13Cr-5Al sample at different conditions. The volume fraction of the GB dislocations increases initially and then becomes approximately constant after 50 ps in the creep process for different temperatures. This variation should correspond to the transition from primary to secondary creep stages. Figure 18 shows that the volume fraction of GB dislocations is above 95% and increases with the increase in temperature because GBs always act as the important sources and sinks of mobile dislocations and a high temperature could enhance the motion of dislocations. The sample snapshots showing the dislocations distribution and defect mesh are also provided in Figure 18. One can find that most of the screw/edge dislocations distribute in GBs and a small amount in grain interiors.

From the viewpoint of material science, dislocations begin to move in modes of slip and climb in high-stress conditions. Dislocation motions may be undermined by GBs and dislocations concentrate at the grain boundary, as presented in Figure 18. As stress increases, the dislocation density decreases because the amount of dislocation emissions is less than that of the dislocation absorption at the grain boundary. From Figure 17 and Figure 18, the dislocation density decreases significantly as the temperature rises. This can be elucidated as follows. The motion of GB dislocations increases with the rising temperature as the diffusion of the GB atoms does. As the temperature increases, the slipped screw dislocations are more likely to be annihilated at the GBs and edge dislocations are absorbed easily in the process of climbing, resulting in a decrease in the density of dislocations.

To evaluate the impact of GS on the dislocation behavior, the variation of the dislocation density with time for Fe-13Cr-5Al samples with different GSs under different stress levels and 1200 K conditions are given and compared in Figure 19. It is revealed that the dislocation density of NC FeCrAl decreases with the increase in the GS because a larger GS model has a lower volume fraction of GB regions. For every model of NC FeCrAl with different GSs, the dislocation density changes little over time under low and medium stress levels, while it decreases gradually with the progress of creep under high stress. It is found that the decrease in the dislocation density with time almost occurs when stress exceeds 1.8 GPa (turning point) for the samples with larger GSs of 18–40 nm at 1200 K. Meanwhile, the turning point of stress is about 1.5 GPa for the model with a 10-nm grain size at the same temperature of 1200 K, as shown in Figure 17d. Thus, we can infer that the smaller the GS is, the smaller the stress at which the dislocation motion begins to affect creep.

### 3.4. Correlation between Creep Deformation and Atomic Diffusion

The apparent increase in the creep rate and the transition of the deformation mechanism from GB diffusion to lattice diffusion with the increasing temperature illustrate that the creep process is thermally activated and the diffusion-driven deformation mechanism is dominant during creep. It is known that creep behavior is predominantly assisted by atomic diffusion, especially at high temperatures. In this section, a detailed comparison of Q_c_ and Q_d_ is made to correlate creep behavior with atomic diffusion.

#### 3.4.1. Activation Energy for Atomic Diffusion Q_d_

Mean square displacement (MSD) provides a quantitative measurement of atom diffusivity and can be calculated by the following equation:(5)$$\mathrm{MSD}=\langle r^{2}\left(t\right)\rangle =\langle \frac{1}{N}\sum_{i=0}^{N}{\left(r_{i}\left(t\right)-r_{i}\left(0\right)\right)}^{2}\rangle$$
where N is the total number of atoms and r_i_(0) and r_i_(t) are the positions of an atom at the initial and current moments, respectively. In this study, an MSD calculation has been conducted to investigate the atom diffusivity of the NC FeCrAl sample under different GS and temperature conditions. The samples were first equilibrated with the NPT ensemble for 300–15,000 ps with zero pressure in three axis directions and further relaxed at a fixed volume to the desired temperature using the NVT ensemble for 300 ps. These simulations were performed using an MD time step of 1 fs. The calculated MSD vs. the time curves obtained from these simulations in the temperature range of 600–1500 K are presented in Figure 20. There exist visible differences in MSD values among Fe, Cr and Al. The obtained MSD values of Fe are much higher compared to those of Cr and Al for a fixed temperature. This result indicates that the diffusivity of Fe in NC FeCrAl is superior to that of Cr and Al. It is also found that the MSD values of each alloying atom increase linearly with time and grow very sharply as the temperature rises.

It is known that a vital correlation between creep behavior and atom diffusion can be identified with the quantitative comparison between Q_c_ and Q_d_. The diffusion coefficient of each alloying atom for NC FeCrAl was computed with Equation (6), based on the calculated MSD values.
(6)MSD=6Dt
where D is the diffusion coefficient and t is the time duration. Furthermore, Q_d_ is obtained from the slope of the logarithmic curves of the diffusion coefficient versus 1/k_B_T.
(7)$$Q_{d}=\frac{\partial lnD}{\partial \left(1/k_{B}T\right)}$$

Figure 21 provides the logarithmic curves of the diffusion coefficient of each alloying atom for the 5.6-nm grain size Fe-13Cr-5Al sample vs. 1/k_B_T at temperatures from 600 to 1500 K. To estimate the variation of Q_d_ with the temperature, all the plots are linearly fitted in three segments. The turning points are about 900 K and 1200 K. It is found that Q_d_ is less than 0.2 eV at a low temperature (LT), between 0.38 and 0.55 eV at a medium temperature (MT) and then increases to 1.0–1.2 eV at a high temperature (HT). It is worth noting that the values of Q_d_ for Fe, Cr and Al are slightly different, although the diffusion coefficients significantly differ from each other. In view of the larger fraction of Fe, it is selected for further analysis on the Q_d_ of NC FeCrAl with different GSs.

Figure 22 presents the logarithmic plots of the diffusion coefficients of Fe vs. 1/k_B_T for the Fe-13Cr-5Al sample with different GSs. Based on the results of Figure 22, the values of Q_d_ for FeCrAl samples with different GSs at different ranges of temperature are summarized in Table 3. It is found that the values of Q_d_ are rarely affected by GS at low temperatures, while GS has a small effect on Q_d_ at temperatures above 900 K. Q_d_ is about 0.21 eV at 600–900 K, between 0.55 and 0.64 eV at a medium temperature and then increases to 0.87–1.20 eV when the temperature is above 1200 K. It is also shown in Figure 22 that the diffusivity of Fe decreases continuously as the GS increases within the present research scope. This could be explained by the fact that, as shown in Table 1, the volume fraction of the grain boundary atoms gradually decreases with the increasing GS.

#### 3.4.2. Activation Energy for Creep Process Q_c_

Figure 23a shows the logarithmic curves of the steady-state creep rate versus 1/k_B_T under different stress conditions. To estimate the variation of Q_c_ with temperature, all the plots are linearly fitted in two segments, taking 1000 K as the turning point. One can find that Q_c_ is less affected by stress at low temperatures than at high temperatures. In the range of the low temperatures, Q_c_ is found to be about 0.1–0.2 eV at different stresses. In contrast, for the higher temperature range, Q_c_ increases roughly from 0.5 to 1.0 eV as stress increases. Figure 23b provides the logarithmic curves for the steady-state creep rate versus 1/k_B_T under different GS conditions at a fixed stress of 0.5 GPa. The plots are linearly fitted in two segments taking 1000 K and 1200 K as the turning point for larger and small GSs, respectively. It is found that the values of Q_c_ rarely change with the GS, other than those that are 5.6 nm at low temperatures. Meanwhile, Q_c_ increases at first and then decreases slightly with the increasing GS at higher temperatures. Based on the results of Figure 23b, the creep activation energies of the FeCrAl samples with different GSs at different ranges of temperature are summarized in Table 4. From Table 4, Q_c_ is about 0.12 eV at the GS of 5.6 nm and about 0.07 eV at larger GSs for the low-temperature range. In contrast, for the higher temperature range, Q_c_ first increases from 0.5 eV to 0.95 eV and then decreases to about 0.8 eV with the increasing GS. The turning point is about 20 nm.

#### 3.4.3. Comparison between Q_c_ and Q_d_

From Table 3 and Table 4, the values of Q_d_ and Q_c_ are deeply influenced by the variation of temperature from 600 to 1500 K. The apparent increase of Q_d_ and Q_c_ with temperature is observed under different GS and stress conditions. In general, the diffusion contribution from atoms within the lattice increases with temperature. Bhatia et al. [32] pointed out that GB atoms dominate the diffusion process at low temperatures, while at temperatures above 0.7 T_m_, the atoms within the grains are more likely to control the diffusion process. Previous studies [56,57] have also shown that the activation energy of lattice diffusion is significantly higher than GB diffusion. In this study, the values of Q_d,LT_ and Q_d,HT_ are assumed to be the activation energies of GB diffusion and lattice diffusion, respectively. As stated in Section 3.3.1, GB and lattice diffusion dominate the creep when stress is below 0.8 GPa. It is shown from Table 3 and Table 4 that the difference between Q_c_ and Q_d_ decreases with the increasing GS and temperature at a fixed stress of 0.5 GPa. There is no significant difference between the two under the larger GSs and higher temperatures. Notably, the values of Q_d,HT_, corresponding to the activation energies of lattice diffusion, are significantly higher than the Q_c,HT_ in small GSs, while there is no great difference between the two in the larger GSs. The Q_c_ of 25 nm NC FeCrAl is about 0.91 eV at temperatures above 1200 K, which is comparable with the Q_d_ of 0.98 eV at the same temperature range. From above, we can believe that the transition of diffusional creep from the GB to the dominant lattice with an increasing temperature and GS exists, which has already been proposed in Section 3.3.1, as predicted by the atomic shear strain ratio in GBs and the grains.

It is also evident from Figure 23a that a variation of stress has an obvious effect on the Q_c_ at a given GS and Q_c_ increases remarkably with stress. The evidence for the transition of the mechanism under low stress has been covered in detail before and will not be repeated here. Under medium stress, the values of Q_c,HT_ are found to exceed the Q_d,LT_ but to be lower compared to the Q_d,HT_, although Q_d,LT_ is slightly higher than Q_c,LT_. It has been reported that the activation energy for the GB slip is lower than lattice diffusion activation energy, but greater than GB diffusion activation energy [58]. Hence, we can believe that the creep behavior is dominated by the GB slip for medium stress, as proposed in Section 3.3.2. Furthermore, under high-stress conditions, the proposed mechanism is the coexistence of the dislocation motion and GB slip. It is known that dislocation motion is composed of diffusion-controlled climb and GB diffusion-controlled glide. Q_c,HT_ is slightly below Q_d,HT_ at higher stresses (2.0 and 2.5 GPa) and Q_c,LT_ is comparable with Q_d,LT_ at the stress of 2.5 GPa. The excellent agreement between Q_c_ and Q_d_ indicates that dislocation motion contributes principally to creep under high stress levels. From above, the transitions of the deformation mechanisms for the creep process of NC FeCrAl proposed in this study are acceptable from a thermodynamic perspective.

## 4. Conclusions

In the present research, the deformation features and the underlying mechanism for the creep process of NC FeCrAl alloys are investigated through MD simulations. The deformation mechanism is firstly discussed in terms of the variations of stress and grain size exponents. Then, the transitions of mechanism with temperature, stress and grain size were further checked from CNA atomic snapshots and other microstructure data statistics of the NC FeCrAl alloys. A vital correlation between the creep process and atomic diffusion is also given to explain the proposed mechanisms. The following is a summary of the major findings:The transitions of mechanisms are mainly controlled by stress. The major mechanism shifts from diffusional creep to GB slip and then to dislocation creep combined with GB slip with stress increasing.Under low-stress conditions, the creep mechanism is dominated by GB diffusion at small GSs, while by lattice diffusion at larger GSs. A high temperature accelerates the transition of the major mechanism from GB diffusion to lattice diffusion.Under medium and high stresses, GB slip together with dislocation motion dominate the overall creep deformation. The amount of GB slip increases with temperature or decreases with the GS. The GS and temperature also have an impact on dislocation behavior. The higher the temperature or the smaller the GS is, the smaller the stress at which the dislocation motion begins to affect creep.The findings of this study offer a comprehensive understanding of the creep behavior of NC FeCrAl alloys at the atomic level, which is hard to obtain from the previous experimental data. This will be conducive not only to the insightful investigation of the creep process in NC materials, but also to the development and design of the nano FeCrAl alloy for its use in engineering.

## Figures and Tables

**Figure 1 nanomaterials-13-00631-f001:**
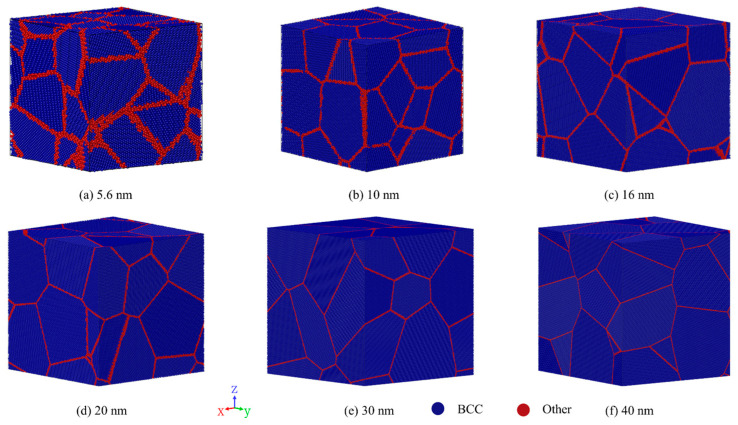
Three-dimensional models for nanocrystalline (NC) FeCrAl samples’ highlighted crystal structure type by common neighbor analysis (CNA) at different grain sizes (**a**) 5.6 nm, (**b**) 10 nm, (**c**)16 nm, (**d**) 20 nm, (**e**) 30 nm, and (**f**) 40 nm.

**Figure 2 nanomaterials-13-00631-f002:**
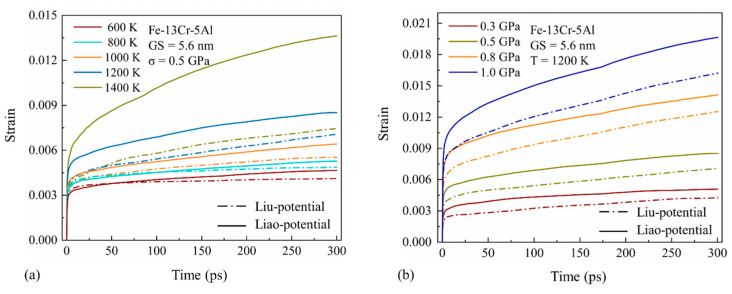
Simulated creep curves for 5.6-nm grain size Fe-13Cr-5Al sample calculated by the Liu-potential and Liao-potential (**a**) under different temperatures and (**b**) stress conditions.

**Figure 3 nanomaterials-13-00631-f003:**
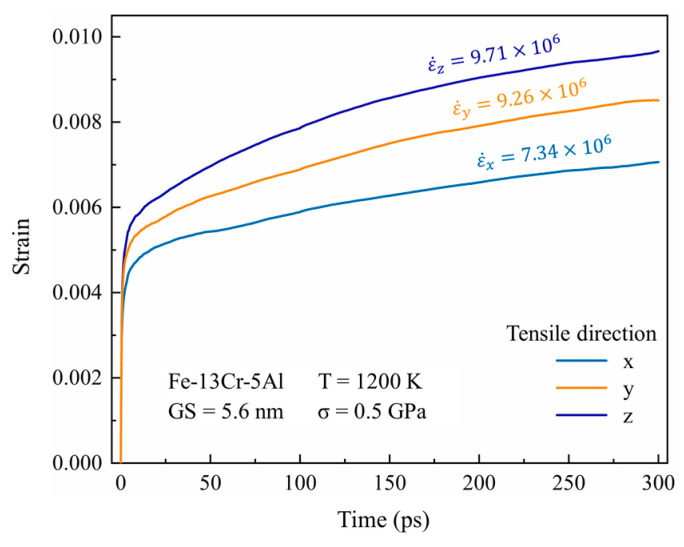
Simulated creep curves for 5.6-nm grain size Fe-13Cr-5Al sample at 1200 K and 0.5 GPa in different tensile directions.

**Figure 4 nanomaterials-13-00631-f004:**
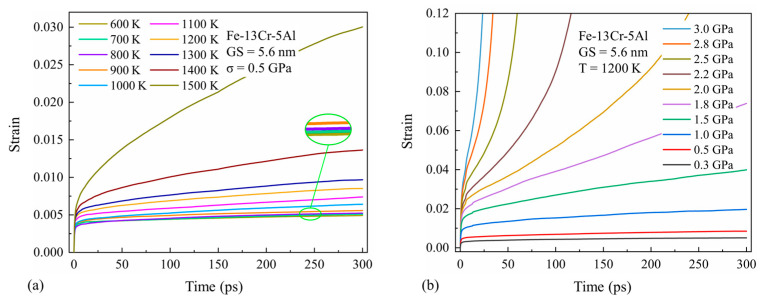
Simulated creep curves for 5.6-nm grain size Fe-13Cr-5Al sample (**a**) under different temperatures and (**b**) stress conditions.

**Figure 5 nanomaterials-13-00631-f005:**
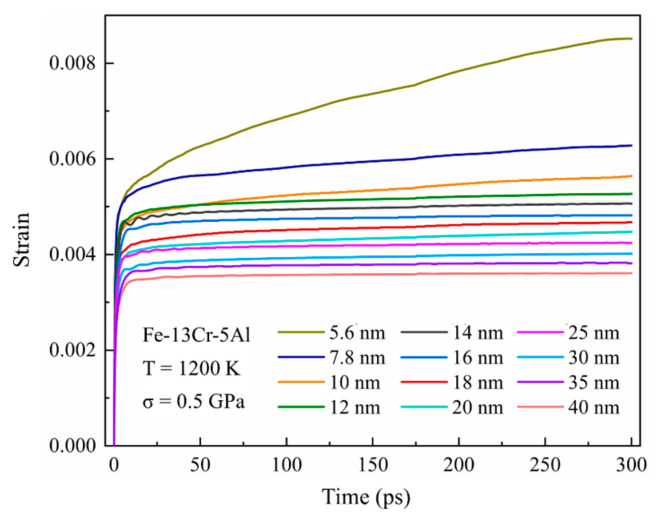
Simulated creep curves for Fe-13Cr-5Al sample with grain sizes from 5.6 to 40 nm at 0.5 GPa for 1200 K.

**Figure 6 nanomaterials-13-00631-f006:**
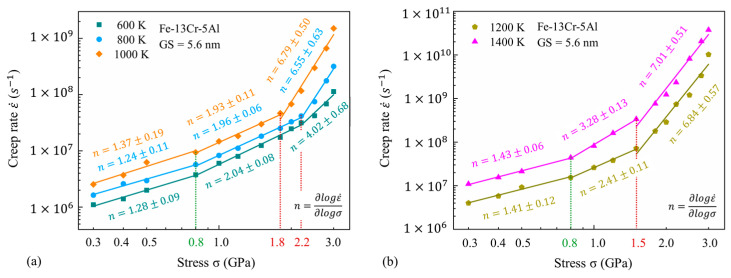
Double-logarithmic curves for steady-state creep rate against stress under different temperatures of 600–1400 K.

**Figure 7 nanomaterials-13-00631-f007:**
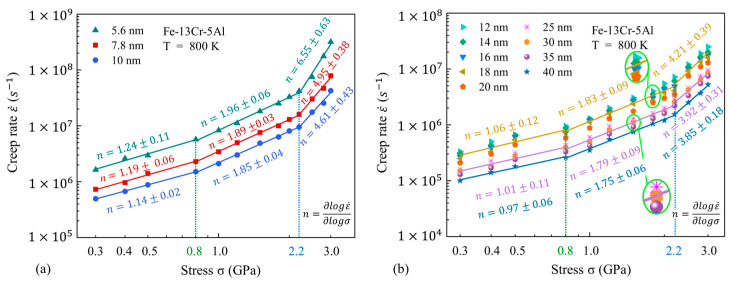
Double-logarithmic curves for steady-state creep rate against stress at different grain sizes (GSs) and a fixed temperature of 800 K.

**Figure 8 nanomaterials-13-00631-f008:**
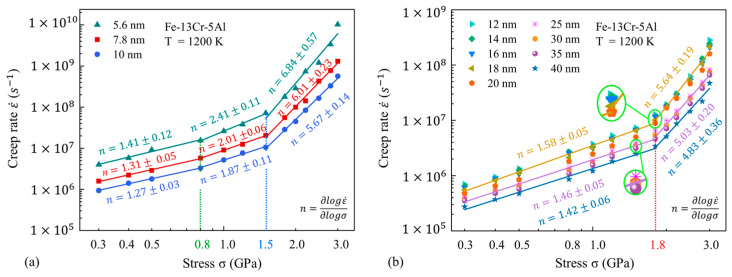
Double-logarithmic curves for steady-state creep rate against stress at different GSs and a fixed temperature of 1200 K.

**Figure 9 nanomaterials-13-00631-f009:**
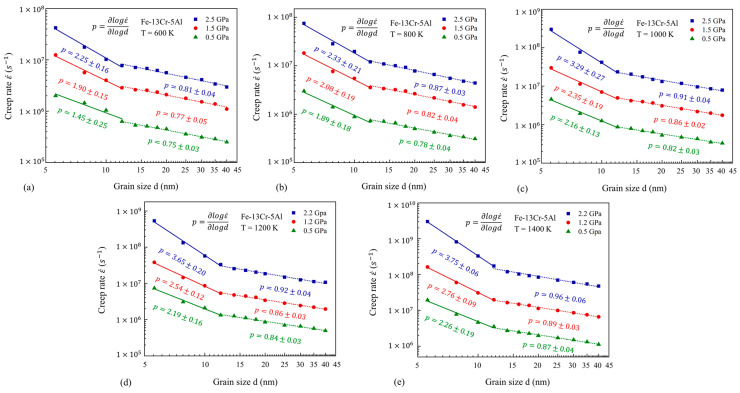
Double-logarithmic curves for steady-state creep rate against grain size under different temperatures of 600–1400 K.

**Figure 10 nanomaterials-13-00631-f010:**
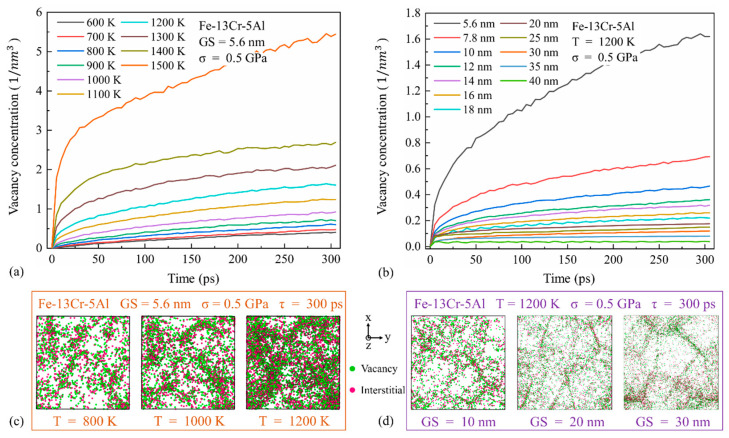
Vacancy concentration changes with time for the Fe-13Cr-5Al sample along with point defect distribution (**a**,**c**) under different temperatures and (**b**,**d**) GS conditions.

**Figure 11 nanomaterials-13-00631-f011:**
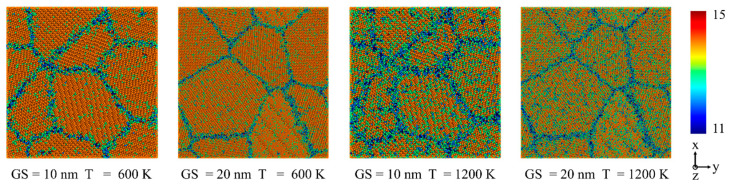
Initial atomic coordination number of NC Fe-13Cr-5Al sample under different temperatures and GS conditions at 0.5 GPa.

**Figure 12 nanomaterials-13-00631-f012:**
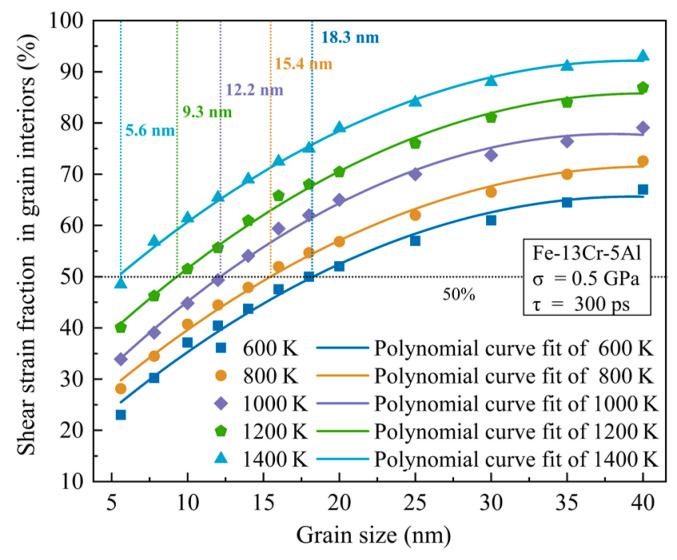
Plots of shear strain fraction in grain interiors vs. grain size for different temperatures and 0.5 GPa at 300 ps.

**Figure 13 nanomaterials-13-00631-f013:**
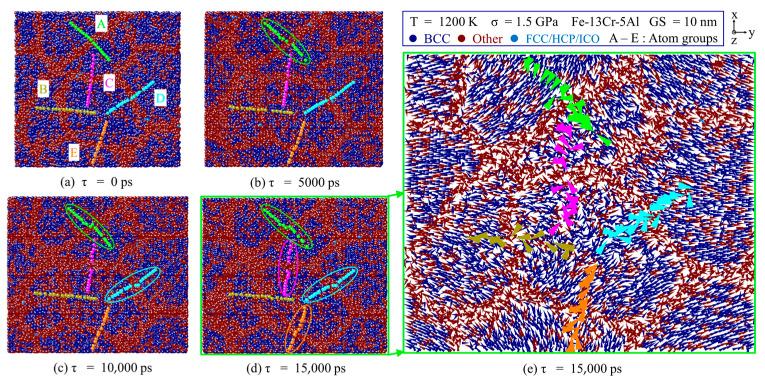
Atomic snapshots colored by crystal structure through CNA at different creep time (**a**) 0 ps, (**b**) 5000 ps, (**c**) 10,000 ps, and (**d**) 15,000 ps at 1200 K and 1.5 GPa. (**e**) The vector displacements of atoms, corresponding to (**d**).

**Figure 14 nanomaterials-13-00631-f014:**
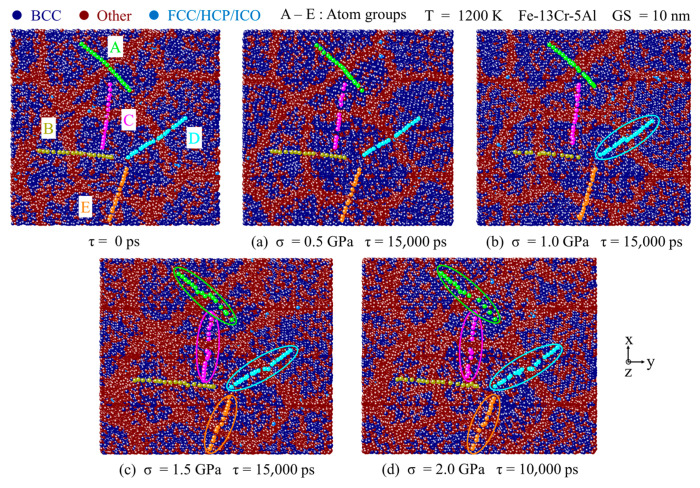
Atomic snapshots colored by crystal structure through CNA under different stress levels (**a**) 0.5 GPa, (**b**) 1.0 GPa, (**c**) 1.5 GPa, and (**d**) 2.0 GPa at 1200 K.

**Figure 15 nanomaterials-13-00631-f015:**
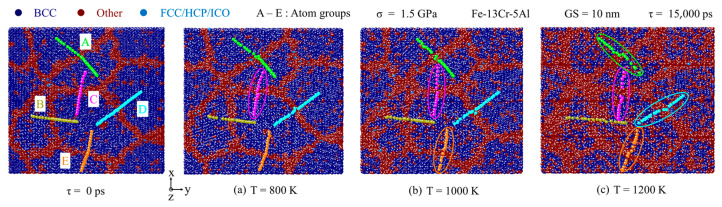
Atomic snapshots colored by crystal structure through CNA at different temperatures from 800 to 1200 K.

**Figure 16 nanomaterials-13-00631-f016:**
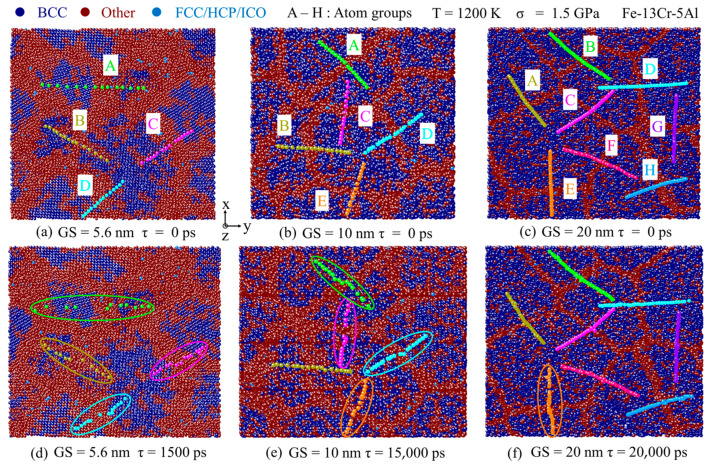
Atomic snapshots colored by crystal structure through CNA at different GSs and 1200 K for 1.5 GPa.

**Figure 17 nanomaterials-13-00631-f017:**
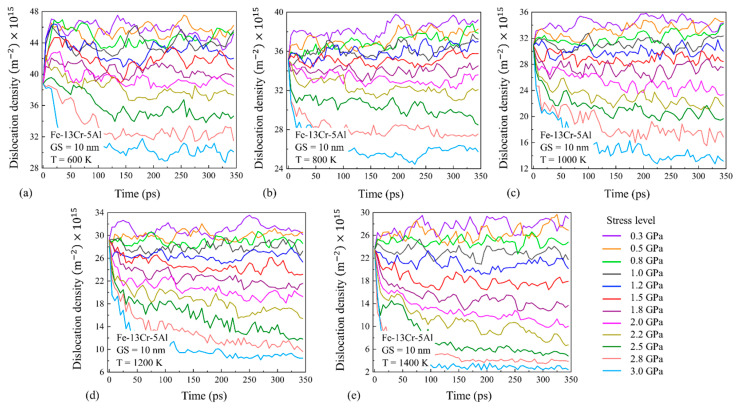
Dislocation density changes with time for 10-nm grain size Fe-13Cr-5Al sample at different stress and temperature conditions.

**Figure 18 nanomaterials-13-00631-f018:**
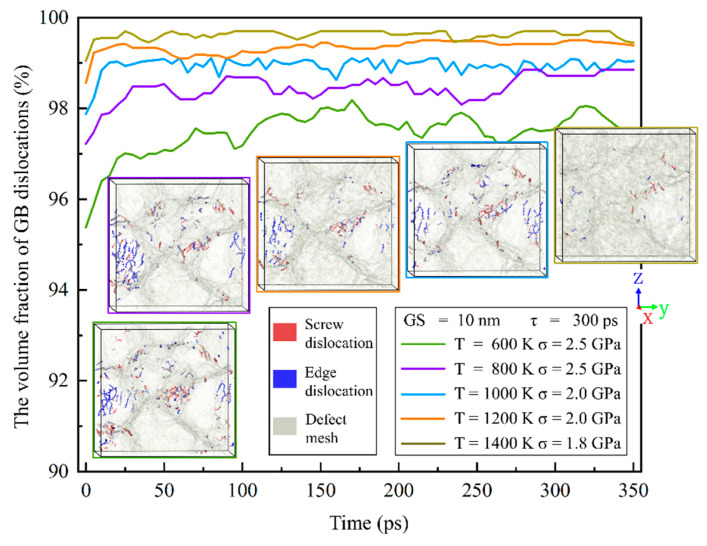
Plots of the volume fraction of GB dislocations vs. time under different conditions along with sample snapshots showing dislocations and defect mesh at 300 ps.

**Figure 19 nanomaterials-13-00631-f019:**
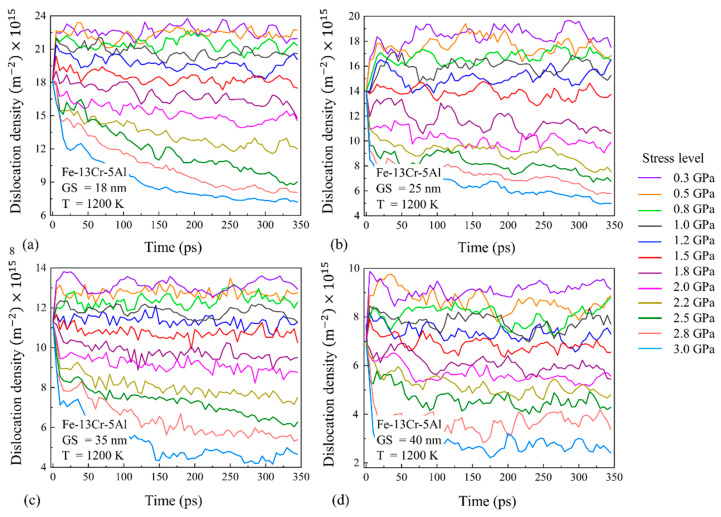
Dislocation density changes with time for different stresses and GSs of (**a**) 18 nm, (**b**) 25 nm, (**c**) 35 nm and (**d**) 40 nm.

**Figure 20 nanomaterials-13-00631-f020:**
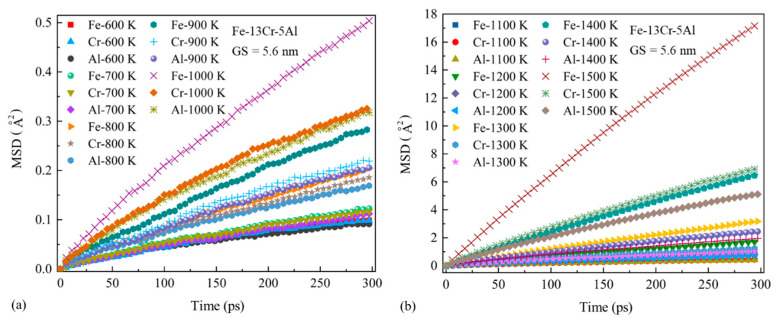
Mean square displacement (MSD) values of Fe, Cr and Al changes with time under different temperatures from 600 to 1500 K.

**Figure 21 nanomaterials-13-00631-f021:**
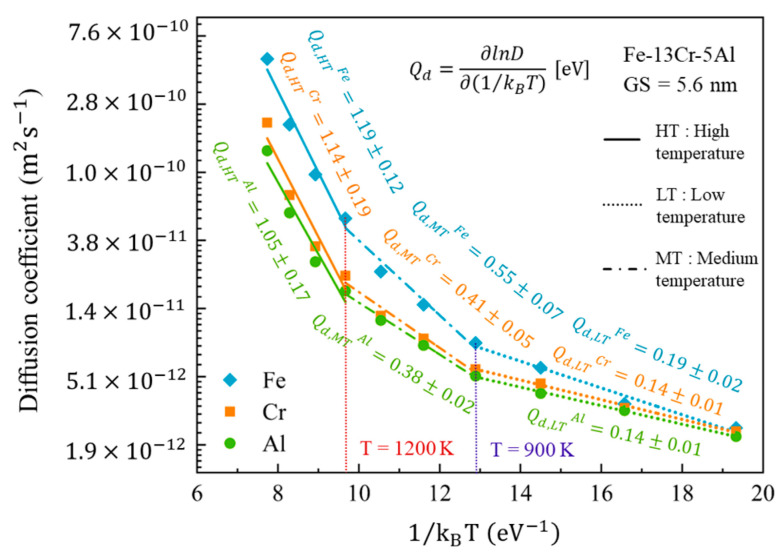
Logarithm diffusion coefficients of Fe, Cr and Al vs. 1/k_B_T in the temperature range of 600 to 1500 K.

**Figure 22 nanomaterials-13-00631-f022:**
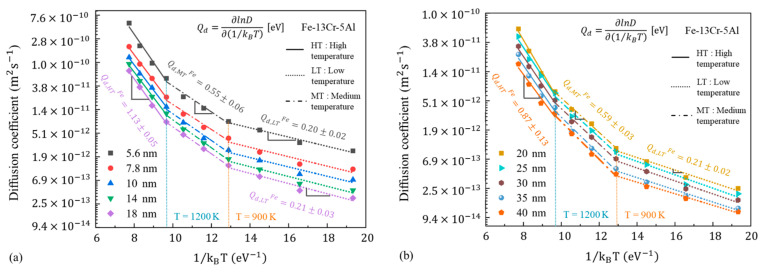
Logarithm diffusion coefficients of Fe vs. 1/k_B_T for Fe-13Cr-5Al sample with different GSs.

**Figure 23 nanomaterials-13-00631-f023:**
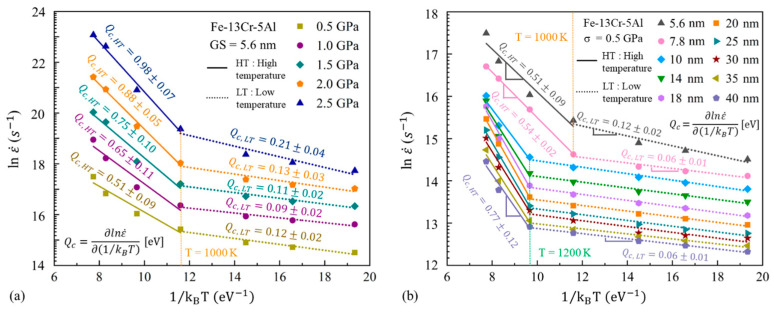
Logarithm steady-state creep rate vs. 1/k_B_T (**a**) at different stress levels, (**b**) under different GS conditions.

**Table 1 nanomaterials-13-00631-t001:** Details of the nanocrystalline (NC) FeCrAl models.

Grain Size (nm)	The Number of Atoms (×10^5^)	Fraction of GB Atoms (%)	Simulation Box Size (nm)	The Number of Grains
5.6	2.4	24.2	14.1	16
7.8	4.8	17.4	17.9	12
10.0	10.1	13.7	22.9	12
12.0	12.2	12.0	24.4	8
14.0	18.6	9.8	28.0	8
16.0	27.8	8.5	32.0	8
18.0	39.6	7.8	36.0	8
20.0	54.3	6.7	40.0	8
25.0	105.6	5.8	50.0	8
30.0	183.2	4.6	60.0	8
35.0	291.3	4.0	70.0	8
40.0	378.1	3.5	76.5	7

**Table 2 nanomaterials-13-00631-t002:** The transition of dominant mechanism with consideration of n and p under different stress, temperature and grain size (GS) conditions.

σ (GPa)		T (K)	GS < 12 nm		GS ≥ 12 nm		Dominant Creep Mechanism
			n	p	n	p	
Low stress	0.3–0.8	600–1400	1.1–1.5	1.4–2.3	~1.0	0.7–0.9	GB diffusion (small GS) Lattice diffusion (larger GS)
Medium stress	0.8–2.2	<800	~2.0	~1.9	~1.8	0.7–0.9	Grain boundary slip (GBS)
	0.8–1.8	800–1000	1.8–2.0	2.0–2.4			
	0.8–1.5	>1000	1.8–3.3	2.5–2.8			
High stress	2.2–3.0	600–1400	4.0–7.0	2.2–3.8	3.8–5.7	0.8–1.0	Dislocation creep + GBS

**Table 3 nanomaterials-13-00631-t003:** Activation energies of diffusion at different temperature levels for Fe-13Cr-5Al sample with different GSs.

Activation Energy of Diffusion (eV)	Grain Size (nm)									
	5.6	7.8	10.0	14.0	18.0	20.0	25.0	30.0	35.0	40.0
Q_d,HT_	1.19	1.09	1.07	1.08	1.13	1.11	0.98	0.96	0.95	0.87
Q_d,MT_	0.55	0.55	0.57	0.58	0.58	0.59	0.60	0.61	0.63	0.64
Q_d,LT_	0.20	0.20	0.20	0.21	0.21	0.21	0.23	0.22	0.21	0.20

**Table 4 nanomaterials-13-00631-t004:** Activation energies of creep at different temperature levels and a fixed stress of 0.5 GPa for Fe-13Cr-5Al sample with different GSs.

Activation Energy of Creep (eV)	Grain Size (nm)									
	5.6	7.8	10.0	14.0	18.0	20.0	25.0	30.0	35.0	40.0
Q_c,HT_	0.51	0.54	0.71	0.84	0.94	0.95	0.91	0.85	0.83	0.77
Q_c,LT_	0.12	0.06	0.08	0.07	0.07	0.07	0.07	0.07	0.06	0.06

## Data Availability

Not applicable.

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
