# Peer review of "Structural Evolution and Transitions of Mechanisms in Creep Deformation of Nanocrystalline FeCrAl Alloys"

_nanomaterials, 2023, doi:10.3390/nano13040631_

Round 1

Reviewer 1 Report

The article is devoted to the study of structural evolution and transitions of mechanisms in creep deformation of nanocrystalline promising fuel cladding material (FeCrAl alloys). Creep is one of important mechanical properties of FeCrAl alloy used as fuel claddings under high temperature condition. This work aims to elucidate the deformation feature and underlying mechanism during creep process using atomistic molecular dynamics simulations. In this regard, the work seems to be relevant and important. Creep mechanisms and the main factors influencing them were studied in detail in this work. A large amount of work has been done.

            There are the following remarks.

  1. Figures 13-16 look great. However, it is desirable to explain the observed effects in more detail. If possible, make estimates of observed shifts, for example, with increasing temperature, based on hypothesized mechanisms (using formula (1)). Then it will be possible to compare the calculated estimates of displacements with those observed in the simulation.
  2. Misprint in line 67: severe
  3. Misprint in line 809: J, W. Steady-State Creep…

The manuscript is recommended for publication with minor correction.

Reviewer 2 Report

Report on the manuscript

 Title:  Structural evolution and transitions of mechanisms in creep deformation of nanocrystalline FeCrAl alloys

 Authors: Huan Yao, Tianzhou Ye, Pengfei Wang, Junmei Wu, Jing Zhang and Ping Chen

 Manuscript ID: nanomaterials-2180957

 I think the readers of this journal will appreciate the results of this manuscript.  Generally speaking, the manuscript is well written, the material is judiciously divided and organized and correct from scientific point of view. Some changes are, however, necessary. For these reasons I can recommend the acceptance of this paper after some corrections presented in the attached file.

Round 2

Reviewer 1 Report

The authors took into account the comments. it is recommended to accept the manuscript in present form.

Reviewer 2 Report

Thanks for your efforts to improve the paper